# Experimental Investigation on Shear Capacity of Steel-Fiber-Reinforced High-Strength Concrete Corbels

**DOI:** 10.3390/ma16083055

**Published:** 2023-04-12

**Authors:** Shu-Shan Li, Jin-Yan Zheng, Jun-Hong Zhang, Hong-Mei Li, Gui-Qiang Guo, Ai-Jiu Chen, Wei Xie

**Affiliations:** 1School of Civil Engineering and Communication, North China University of Water Resources and Electric Power, Zhengzhou 450046, China; lishushan@ncwu.edu.cn (S.-S.L.); zhengjinyan@stu.ncwu.edu.cn (J.-Y.Z.);; 2Engineering Technology Research Center for Structural Vibration Control and Health Monitoring of Henan Province, Zhengzhou 450046, China

**Keywords:** SFRHSC, corbel, small shear span/depth ratio, shear capacity, strut-and-tie model

## Abstract

As short cantilever members, corbels are mainly used to transfer eccentric loads to columns. Because of the discontinuity of load and geometric structure, corbels cannot be analyzed and designed using the method based on beam theory. Nine steel-fiber-reinforced high-strength concrete (SFRHSC) corbels were tested. The width of the corbels was 200 mm, the cross-section height of the corbel column was 450 mm, and the cantilever end height was 200 mm. The shear span/depth ratios considered were 0.2, 0.3, and 0.4; the longitudinal reinforcement ratios were 0.55%, 0.75%, and 0.98%; the stirrup reinforcement ratios were 0.39%, 0.52%, and 0.785%; and the steel fiber volume ratios were 0, 0.75%, and 1.5%. According to the test results, this paper discusses the failure process and failure mode of corbel specimens with a small shear span/depth ratio and analyzes the effects of variables such as shear span/depth ratio, longitudinal reinforcement ratio, stirrup reinforcement ratio, and steel fiber volume content on the shear capacity of corbels. The shear capacity of corbels is significantly affected by the shear span/depth ratio, followed by the longitudinal reinforcement ratio and the stirrup reinforcement ratio. Moreover, it is shown that steel fibers have little impact on the failure mode and ultimate load of corbels, but can enhance the crack resistance of corbels. In addition, the bearing capacities of these corbels were calculated by Chinese code GB 50010-2010 and further compared with ACI 318-19 code, EN 1992-1-1:2004 code, and CSA A23.3-19 code, which adopt the strut-and-tie model. The results indicate that the calculation results by the empirical formula in the Chinese code are close to the corresponding test results, while the calculation method based on the strut-and-tie model of a clear mechanical concept yields conservative results, and hence the related parameter values must be further modified.

## 1. Introduction

As important supporting members of engineering structures, such as bridges, factory building, and gate piers, corbels are expected to possess high crack resistance and bearing capacity [1]. The internal stress state of corbels is complicated under load, where the strain is nonlinear and the stress is significantly redistributed, features which are not appropriate for assumed design theory [2,3]. Corbels are classified as members of a discontinuity region. Steel fiber high-strength concrete (SFHSC) is a new type of composite material formed by adding steel fiber to high-strength concrete. The addition of steel fibers helps to enhance the concrete’s ability to resist cracking and deformation under various types of loading, which is important for ensuring the structural stability and durability of the concrete element. It can effectively improve the structural crack resistance and shear capacity of corbels and ameliorate the reinforcement density and hence improve the seismic performance of whole structures [4,5]. Thomas et al. [6] investigated the impact of the steel fiber mixing range on the seismic performance of beam–column joints. They discovered that structures employing fiber concrete exhibited a greater bearing capacity and energy dissipation capacity. Ghassan et al. [7] investigated a new type of reinforcing to greatly improve the ductility and strength of SSWs. Zhu et al. [8] observed that the addition of steel fiber to concrete resulted in a higher ultimate compressive strain, ultimately leading to an enhancement in the ductility and flexural strength of beams constructed from fiber-reinforced concrete.

At present, the calculation formulas of corbel bearing capacity in many design codes are mainly empirical formulas, which are obtained by statistical regression analysis based on a great deal of experimental data [9]. In recent years, the simplified strut-and-tie model (STM) has been recommended in the appendix of American ACI 318-19 code for corbel bearing capacity design [10]; there are similar provisions in European EN 1992-1-1:2004 code [11] and Canadian code CSA A23.3-19 code [12]. In these codes, STM is defined as a special application of the plastic lower bound principle in practical structural design, which is specified to be used in the case of non-linear distribution of section strain. Because STM only needs to satisfy the equilibrium condition and yield criterion and does not consider the strain coordination in solid mechanics, STM is widely utilized to study the shear capacity of reinforced concrete deep beams, beam column joints, corbels, and other components [13,14,15,16,17,18].

Many studies have been conducted to define the mechanical properties of corbels through experimental and theoretical research. Fattuhi [19,20] analyzed the bearing capacity of corbels with a bending compression model and a truss model. It was shown that the bending compression model is relatively accurate in predicting the bearing capacity of corbels with bending compression failure, while the truss model can predict the bearing capacity of corbels with any failure form. According to the experiments of 16 corbel specimens, Yong et al. [21] examined the influences of horizontal load, reinforcement ratio, and shear span/depth ratio on the bearing capacity of corbels. Their findings showed that the bearing capacity of the corbels increased with higher concrete strength when subjected to these variables. Foster et al. [22] indicated that the cracking and bearing capacities of corbels are significantly affected by the shear span/depth ratio, and as the shear span/depth ratio increases, the bearing capacity decreases. In particular, concrete strength and corbel height are also important factors affecting the cracking and bearing capacity of corbels. To a certain extent, the stirrup affects the failure form of corbels and increasing the stirrup reinforcement ratio can improve the bearing capacity of corbels. Note that the transverse strain of concrete is very disadvantageous to the function of the concrete compression bar, while the stirrup restrains the concrete compression bar and improves the strength of the concrete compression bar. Hwang et al. [23] proposed a softened STM to determine the shear capacity of corbels. The shear capacities predicted by the softened STM and by the empirical formula of ACI 318 were further compared with the experimental data of 178 corbels collected. The results show that the calculation results by the softened STM are in good agreement with the experimental data, whereas those obtained using ACI 318 are more conservative. Through superimposing the shear capacity contribution of STM provided by concrete cracking and main reinforcement with that provided by stirrup, Russo et al. [24] put forward a new model to determine the shear bearing capacity of reinforced concrete corbel, which demonstrated a better fitting of corbel specimen shear capacity compared to the formula in the ACI code. Campione et al. [25] conducted an experimental investigation on the bending performance of corbels and proposed a calculation method for determining the shear capacity of corbels reinforced with stirrups. Similarly, Gao et al. [26] discussed the mechanical properties and failure characteristics of steel-fiber-reinforced concrete corbels, and showed that the addition of steel fiber increases the cracking load and ultimate load of corbels. Furthermore, Gao et al. [27,28] carried out experimental research on steel-fiber-reinforced concrete corbel specimens, and derived the calculation formula of the shear capacity of steel-fiber-reinforced concrete corbel. Chen et al. [29] examined the influences of shear span/depth ratio and concrete strength on the bearing capacity of corbels. In the case of small shear span/depth ratio, the failure modes of corbels can be divided into shear failure and diagonal compression failure. The shear capacity of corbels basically increases linearly with the increase in concrete strength, yet obviously decreases with the increase in shear span/depth ratio. Mehmet et al. [30] found that steel fiber concrete effectively delayed the destruction of corbel by adding steel fiber to the corbel, and with greater steel fiber content, the bearing capacity of the steel fiber concrete corbel was stronger. Saddam Kha et al. [31] conducted an experimental study on the shear performance of steel fiber concrete corbels, and the results showed that the presence of fiber improved the shear strength and deformation ability of the components.

In the existing research, an ordinary-strength concrete corbel is mostly taken as the research object. In addition, given the complex mechanical performance of corbels and the discreteness of steel fiber concrete materials, there is still disagreement on the transfer mechanisms and influencing parameters. Therefore, the calculation model of corbel bearing capacity needs to be further improved. It is necessary to conduct experimental research and theoretical analysis on the shear behavior of SFRHSC corbels and to examine the reliability of the calculation method of corbel bearing capacity in the current code, and put forward a reasonable design calculation method, which has important theoretical significance and practical value. To this end, this paper investigates the failure process and failure mode of corbels according to experiments using nine SFRHSC corbels of C60 concrete. In particular, this paper examines the influences of several test parameters on the normal or inclined section cracking load and ultimate load of corbels and focuses on the calculation methods of the shear capacity of corbels.

## 2. Test Overview

### 2.1. Specimen Design

The corbel specimens with double-sided support are shown in Figure 1. The width of the corbels is 200 mm, the cross-section height of the corbel column is 450 mm, and the cantilever end height is 200 mm. As shown in Table 1, the shear span/depth ratios considered are 0.2, 0.3, and 0.4; the longitudinal reinforcement ratios are 0.55%, 0.75%, and 0.98%; the stirrup reinforcement ratios are 0.39%, 0.52%, and 0.785%; and the steel fiber volume ratios are 0, 0.75%, and 1.5%. This paper analyzes the influences of these factors on the failure mode and ultimate bearing capacity of corbels, and also discusses the shear transfer mechanism of corbels with a small shear span/depth ratio.

### 2.2. Test Setup and Instrumentation

As shown in Figure 2a, in order to observe the strain changes in the concrete surface in the normal or inclined section of corbels and to control the test process, 16 concrete strain gauges were symmetrically arranged on the possible cracks on the front of each corbel specimen, with 8 strain gauges in the normal section and 8 strain gauges in the inclined section. To accurately measure the stress state of reinforcement, strain gauges were installed in the corresponding positions of longitudinal reinforcement and horizontal reinforcement. The steel strain gauges were connected as shown in Figure 2b. The first number in the strain gauge number represents the layers of reinforcement, and the second represents the specific position of the reinforcement strain gauge.

In addition, three displacement meters were arranged on the back of the corbel specimen, shown in Figure 3, to measure the mid-span displacement of the corbel specimen during loading. The Isolated Measurement Pods (IMP) data automatic acquisition system was used to collect data on concrete strain, reinforcement strain, and displacement. Additionally, the crack width of concrete was measured with a KON-FK (N) crack observation instrument (Beijing Concrete Engineering Testing Technology Co., Ltd., Beijing, China) with a minimum scale of 0.01 mm, and the status of the crack propagation was simultaneously recorded.

Figure 4 shows the loading system utilized, namely the YA-5000 pressure testing machine with load capacity of 5000 kN. Firstly, the corbel specimens were preloaded to 30 kN to eliminate the sand cushion and other gaps. After the preloading was completed and the data were cleared, the corbel specimens were loaded formally with a loading speed of 1 kN/s. The load was increased by 60 kN per level prior to cracking and by 100 kN per level after cracking. The load was maintained for 2 min for each load level. After the reading was stable, the data were recorded. When the specimen is about to show normal cracking, diagonal cracking, and failure state, the loading rate should be reduced appropriately [32,33]. After the failure of the specimens, the pressure testing machine unloaded to 70% of the ultimate load at a constant speed, so as to observe the ultimate failure mode of the corbel more in depth.

### 2.3. Material Properties

The design strength of the matrix of concrete was C60 according to the “Technical Regulations for Fiber Concrete Structure” [34]. The mixed proportion of steel fiber concrete is shown in Table 2. The length of steel fiber was equal to 32 mm, the diameter was 0.75 mm, the aspect ratio was 42.7 mm, and the tensile strength was more than 600 MPa. The mechanical properties of steel fiber concrete are listed in Table 3.

## 3. Test Discussion and Results

### 3.1. Failure Mode of Corbel with Small Shear Span/Depth Ratio

The corbel member has a relatively small shear span/depth ratio, resulting in a failure mode where cracks form along the connection line between the inner side of the loading plate and the lower cylinder. However, when the inclined crack penetrates into the connection surface of the corbel and the column, it will extend downward along the direction parallel to the connection surface. A penetrating inclined crack can be seen at the connection surface of the loading plate and column when the corbel fails; at the same time, the longitudinal tensile steel bar at the top reaches the yield strength. This is a kind of concrete shear failure resulting from shear force; its failure section is nearly vertical and its cracks are slightly inclined shear cracks. Figure 5a displays the diagonal shear failure pattern of a typical corbel specimen. 

When the cracking load of the inclined section is reached, the middle part of the diagonal strut is perpendicular to the direction of the main tensile stress, and a number of abdominal shear oblique cracks roughly parallel to each other appear successively, and oblique prisms appear in the belly of the corbel. With the increase in load, a main oblique crack appears along the boundary line between the inner side of the loading plate and the lower cylinder. After the appearance of the main diagonal crack, the longitudinal tensile reinforcement reaches the yield strength when approaching the ultimate load, and the width of the main diagonal crack reaches about 0.5 mm. The compressive stress of the concrete diagonal bar gradually increases until the compressive strength of the prism is reached. Figure 5b displays the diagonal compression failure pattern of a typical corbel specimen.

### 3.2. Failure Process of Corbel with Small Shear Span/Depth Ratio

From the initial loading to failure, corbels with small shear span/depth ratios generally experiences three stages: cracking, critical cracking, and failure. From initial loading to cracking, the corbel is in the elastic working stage and the surface strain of concrete presents a linear distribution. With the increase in load, vertical normal cracks appear at the junction of the corbel and upper cylinder, and the normal crack load is 20~30% of the ultimate load. Then, inclined cracks appear under the inner side of the loading plate, and the corresponding inclined crack load is 30~50% of the ultimate load. With the increase in vertical load, the inclined cracks develop downwards roughly along the inner side of the loading plate and point to the junction of the lower cylinder. The normal crack generally can extend about 10~20 cm, while the inclined crack extends downwards rapidly with the onset of the initial crack and the crack width increases with the increasing load. When the load value increases to 80~90% of the ultimate load, the longitudinal tensile reinforcements of corbel yield, the diagonal crack suddenly expands. Finally, the corbel fails when it reaches the shear strength limit. The crack patterns of the specimens when they ultimately reach failure are shown in Figure 6.

### 3.3. Concrete Strain of Corbels Normal Section and Diagonal Section

Figure 7a shows the strain distribution of a typical corbel normal section. At the initial stage, the normal section strain shows a linear relationship with the height, approximately following the plane section assumption. While approaching the normal section cracking load, the normal section strain does not conform to the assumption of plane sections.

Figure 7b displays the concrete strain distribution of the corbel diagonal section. Before cracking, the strain gradually decreases along the upper part to the lower part of the diagonal section and generally increases with the increase in load. The main tensile stress on the diagonal section is approximately perpendicular to the connecting line between the center of the loading support and the lower column, which is consistent with the case of ordinary reinforced concrete corbel [24]. It follows that the steel fiber does not change the direction of principal tensile stress in high-strength concrete. However, before the normal section is about to crack, the strain on the inclined section sometimes decreases or even becomes compressed, which indicates that the internal stress of the SFRHSC diagonal compression bar is very complex.

### 3.4. Strain of Longitudinal Bars in Corbel

Figure 8 shows the relationship between the load and average strain of longitudinal reinforcement of corbel specimens with different shear span/depth ratios. It can be seen from Figure 6 that under 20~30% of the ultimate load, the stress of reinforcement is only about 15% of the yield strength, indicating that the load is mainly borne by the high-strength concrete. After the corbel normal section cracked, the strain of the steel bar abruptly increased, which shows the longitudinal tensile reinforcement will bear much more load. The longitudinal reinforcement plays an essential role in the shear bearing capital of corbels [35]. In addition, all longitudinal bars at the failure end reach the yield strength, and the smaller the shear span/depth ratio, the greater the load value. It can also be observed that the average strain of longitudinal reinforcement decreases with the decrease in shear span/depth ratio, indicating that the bending action of longitudinal reinforcement is decreasing.

Figure 9 shows the measured load–strain curves of corbel specimens with different steel fiber contents. Before the cracking, the stress of the steel bar is considerably small and increases approximately linearly with load. That is to say, the deformation of the corbel is very small, and the effect of the steel bar is not obvious. After the cracks appear, the stress of the steel bar increases rapidly and the curve becomes flat, showing that the growth rate of the stress of the steel bar is obviously faster than the change rate of load. After cracking, the strain growth rate of the main reinforcement of MC08 corbel without steel fiber is obviously higher than that of the corbels with steel fiber, which indicates that the steel fiber in concrete also bears part of the external load. Close to the failure of MC08 and MC09 corbels, all the longitudinal reinforcements have yielded. Moreover, some stirrups reach the yield strength when the corbels are damaged, which indicates that stirrups also play an important role in the shear resistance of steel-fiber-reinforced high-strength concrete corbels [36].

## 4. Influencing Factors of Shear Capacity of Corbels

The shear capacity of steel-fiber-reinforced concrete corbels is influenced by several interrelated factors, including the steel fiber content ratio, shear span/depth ratio, longitudinal reinforcement ratio, concrete strength, and stirrup reinforcement ratio. This study primarily focuses on analyzing the impact of the shear span/depth ratio, longitudinal reinforcement ratio, stirrup reinforcement ratio, and steel fiber content ratio.

### 4.1. Shear Span/Depth Ratio

Three specimens, MC01 (*λ* = 0.2), MC02 (*λ* = 0.3), and MC03 (*λ* = 0.4), were utilized to explore the effect of shear span/depth ratio on the mechanical behavior of corbels. Figure 10 shows the relationships between the shear span/depth ratio of corbel specimens and their cracking load and ultimate load. The normal section cracking load is defined as the load when the first vertical flexible crack occurs, and the diagonal cracking load is defined as the load when the first diagonal crack occurs. It can be seen from Figure 8 that the normal section cracking load decreases with increasing shear span/depth ratio. Particularly, the normal section cracking loads of MC02 and MC03 decrease by 18.5% and 38.3%, respectively, compared with that of MC01. Moreover, the diagonal cracking loads of MC02 and MC03 decrease by 27.5% and 36.3%, respectively. The effect of the shear span/depth ratio on the cracking load of the corbel’s inclined section can be explained as follows. The direction of main tensile stress in the corbel is approximately perpendicular to the line connecting the center of loading support and the lower column, and the vertical compressive stress near the support reduces the main tensile stress. With the decrease in shear span/depth ratio, the vertical compressive stress increases and the cracking load of the corbel’s inclined section increases correspondingly. In addition, it can be seen from Figure 8 that the ultimate load of the corbel exhibits a nearly linear reduction trend as the shear span/depth ratio increases. The ultimate loads of MC02 and MC03 decrease by 6.5% and 20.9%, respectively, compared with that of MC01.

### 4.2. Longitudinal Reinforcement Ratio

Figure 11 displays the relationship between the cracking or ultimate loads and longitudinal reinforcement ratios of corbel specimens MC02 (*ρ*_s_ = 0.55%), MC04 (*ρ*_s_ = 0.75%), and MC05 (*ρ*_s_ = 0.98%). It can be seen that the normal section cracking load of corbel changes slightly and even decreases with the increase in longitudinal reinforcement ratio. In general, the normal section cracking load of the corbel is a negligible effect by the longitudinal reinforcement ratio. As the longitudinal reinforcement ratio increases, the cracking load of the corbel’s inclined section increases. The cracking loads of specimens MC04 and MC05 increase by 22.4% and 33.3%, respectively, compared with that of specimen MC02. Similarly, the increase in longitudinal reinforcement ratio also leads to the increase in ultimate load. The ultimate loads of specimens MC04 and MC05 increase by 6.3% and 19.4% when the longitudinal reinforcement ratios increase from 0.55% to 0.75% and 0.98%, respectively. Generally, the influence of the longitudinal reinforcement ratio on the ultimate load is less significant compared to the effect of the shear span/depth ratio.

### 4.3. Stirrup Ratio

According to the test results of specimens MC02 (*ρ*_sv_ = 0.785%), MC06 (*ρ*_sv_ = 0.39%), and MC07 (*ρ*_sv_ = 0.52%), the influence of stirrup ratio on the cracking load and ultimate load of corbels is shown in Figure 12. It can be seen that a slight effect is imposed on the cracking load of the normal section by the increase in stirrup ratio. When the stirrup ratio increases from 0.39% to 0.52% and 0.785%, the cracking load of MC07 decreases by 14.3%, while that of specimen MC02 increases by 14.8%. The diagonal cracking load of the corbel specimen shows an increase with an increase in the stirrup ratio: the diagonal cracking loads of specimens MC07 and MC02 increase by 2.3% and 23.7%, respectively, compared with that of specimen MC06. This indicates that stirrups can effectively restrain the transverse strain of concrete in the diagonal compression bar and improve the diagonal cracking load of corbels. Similarly, the ultimate load of the corbel increases as the stirrup ratio is increased. The ultimate loads of specimens MC07 and MC02 increase by 12% and 22.6%, respectively, compared with specimen MC06, indicating that stirrups can effectively restrain the transverse strain in the concrete diagonal column and hence improve the ultimate load of the diagonal column in the corbel specimen.

### 4.4. Steel Fiber Content

According to the test results of corbel specimens MC02 (*ρ*_f_ = 1.5%), MC08 (*ρ*_f_ = 0), and MC09 (*ρ*_f_ = 0.75%), Figure 13 shows the influence of steel fiber content on the cracking load and ultimate load of corbels. The cracking load of the corbel’s normal section increases with the increase in steel fiber content: the cracking loads of MC09 and MC02 increase by 20% and 46.7%, respectively, compared with that of MC08. Further, with the increase in the steel fiber content, the diagonal cracking load of MC09 is 15.1% higher than that of MC08, while that of MC02 is 12.4% higher. When the steel fiber content exceeds 0.75%, the increase in steel fiber content has no significant impact on the diagonal cracking load of steel-fiber-reinforced high-strength concrete corbels, which is consistent with the conclusion in the literature [37]. Finally, compared with specimen MC08, the ultimate load of specimen MC09 is reduced by 1%, and the ultimate load of specimen MC02 is increased by 6%.

## 5. Calculation Method of Shear Capacity of Corbel

The calculation method of corbel shear capacity in Chinese GB 50010-2010 is a semi-empirical and semi-theoretical calculation formula that is based on a substantial number of tests. Herein, the corbel is simplified as a truss model in terms of its mechanical characteristics. The distribution of tensile stress in the longitudinal main reinforcement is uniform, which is equivalent to the tension bar in the truss model. An oblique pressure band is formed between the inclined cracks on both sides of the loading support, and the distribution of compressive stress in the oblique pressure band is uniform, which is equivalent to the compression bar in the truss model. During failure, the longitudinal bars yield first, and then the concrete in the diagonal compression bar reaches its ultimate compressive strength, which often has the characteristics of diagonal compression failure. After the yield of longitudinal bars, the concrete in the diagonal compression bar bears great compressive stress and tensile stress, and the direction of compressive stress is parallel to that of the diagonal compression bar, while the existence of tensile stress reduces the compressive strength of concrete. Because of these mechanical characteristics, the main problems of corbel design are the cross-section size of the diagonal strut related to the huge shear force, and the top horizontal stressed reinforcement and horizontal stirrup related to the horizontal tie rod.

For the corbel with a small shear span/depth ratio, ACI 318-19 Code [7], EN 1992-1-1:2004 Code [8], and CSA A23.3-19 Code [9] provide the corresponding calculation method of the shear capacity based on the strut-and-tie model (Figure 14), and the determined shear capacity calculation model is in good agreement with the actual situation. The strut-and-tie model is a mechanical model abstracted from the structural entity. Under the action of load and support reaction, the strut-and-tie model is in a state of force balance. The strut, tie rod, and node area of the strut-and-tie model have limited width, which should be considered when selecting the strut-and-tie model.

### 5.1. GB 50010-2010 Code

According to the provisions of GB 50010-2010 [9], under the action of vertical concentrated load, the shear capacity in the diagonal section of a corbel specimen shall meet the following requirements:(1)$$V=0.85f_{y}h_{0}A_{s}/a$$
where *h*_0_ is the effective height of corbel section, *A*_s_ is the section area of longitudinal reinforcement, and *a* is the horizontal distance from the load point to the edge of lower column.

### 5.2. ACI 318-19 Code

It can be seen from Figure 12 that the shear capacity of a corbel specimen can be calculated by the following formula:(2)$$V=F_{\mathrm{ns}}\sin\theta$$
(3)$$F_{\mathrm{ns}}=f_{\mathrm{ce}}A_{\mathrm{cs}}$$
(4)$$A_{\mathrm{cs}}=bw_{s}$$
(5)$$f_{\mathrm{ce}}=0.85\beta_{s}\beta_{c}{f^{\prime }}_{c}$$
where *F*_ns_ is the axial bearing capacity of the compression bar, *A*_cs_ is the cross-sectional area of the compression bar, *b* is the thickness of the corbel, *w*_s_ is the width of the compression bar, *f*_ce_ is the effective compressive strength of the concrete, *β*_s_ is the influence coefficient of the effective compressive strength of the compression bar (*β*_s_ = 1.0 for the compression bar with equal cross-section [10]), and *f*_c_’ is the standard cylinder compressive strength of concrete.

### 5.3. EN 1992-1-1:2004 Code

The calculation process for the strut-and-tie model utilized in EN 1992-1-1:2004 [8] is basically the same as that in ACI 318-19 [11], but with a different correction for *f*_ce_ in the compression bar. In fact, the strut is generally designed as a bottle-shaped compression bar, and is usually simplified as an equal cross-section compression bar, sometimes as a uniform variable cross-section compression bar. The relationship between compressive strength and principal tensile strain is given in EN 1992-1-1:2004 [8].

Without transverse tensile stress or with compressive stress,
(6)$$\sigma_{\mathrm{Rd},\max}=f_{\mathrm{cd}}$$

When there is transverse tensile stress and cracking is allowed,
(7)$$\sigma_{\mathrm{Rd},\max}=0.6\nu f_{\mathrm{cd}}$$
where *σ*_Rd,max_ is the effective compressive strength of concrete in the compression bar, *f*_cd_ = *α*_cc_*f*_ck_/*γ*_c_ is the compressive strength value of concrete (*α*_cc_ is the coefficient considering the adverse effect of long-term effect and load type on compressive strength, the recommended value in code is 0.85, *f*_ck_ is the standard value of concrete compressive strength; the sub-coefficient of concrete *γ*_c_ is 1.5 in persistent state and 1.2 in accidental state), and *ν* = 1 − *f*_ck_/250 is the strength reduction factor of concrete after cracking under shear force (herein, *ν* is taken as 1.0).

### 5.4. CSA A23.3-19 Code

According to CSA A23.3-19 [12], the bearing capacity of a corbel can be calculated by the following formula:(8)$$F_{\mathrm{ns}}=\phi_{c}f_{\mathrm{cu}}A_{\mathrm{cs}}$$
(9)$$f_{\mathrm{cu}}=\frac{f_{c}{}^{\prime }}{0.8+170\varepsilon_{1}}\leq 0.85f_{c}{}^{\prime }$$
(10)$$\varepsilon_{1}=\varepsilon_{s}+(\varepsilon_{s}+0.002)\cot^{2}\theta_{s}$$
where *ϕ*_c_ is the resistance coefficient of concrete, taken as 1; *f*_cu_ is the compressive strength of concrete cube; *θ*_s_ is the minimum angle between the compression bar and its adjacent pull rod; *ε*_s_, ranging from 0.0012 to 0.0038, is the tensile strain in the pull rod with the inclination angle *θ*_s_ of the compression bar; and *f*_c_’ is the compressive strength of concrete cylinder.

### 5.5. Comparative Analysis of Test Results

The above-mentioned national codes were utilized to calculate the bearing capacity of nine specimens. Table 4 presents the calculated values by these codes and the corresponding experimental values. The table shows that the average ratios of the test value to the calculated values by GB 50010-2010, ACI 318-19, EN 1992-1-1:2004, and CSA A23.3-19 are 1.511, 1.900, 1.810, and 1.928, respectively, and the corresponding variances are 0.055, 0.038, 0.035, and 0.038. Generally, all the calculated results are lower than the test values. The calculation results by GB 50010-2010 are relatively close to the test results, but the variance is larger than the other cases. In contrast, the results from the strut-and-tie model with a clear mechanical concept are more conservative yet with smaller variance. At present, the European, American, and Canadian codes take the strut-and-tie model as an alternative design scheme. However, these results indicate that the specific parameters need to be further studied.

## 6. Conclusions

To study the failure mode of corbels and their shear bearing capacity, nine C60 steel-fiber-reinforced high-strength concrete corbels were tested. We evaluated GB 50010-2010, ACI 318-19, EN 1992-1-1:2004, and CSA A23.3-19 specifications based on the test results, which helped us identify their limitations. The following conclusions can be made:(1)Steel-fiber-reinforced high-strength concrete corbel specimens with a shear span/depth ratio ranging from 0.2 to 0.4 exhibit two typical failure modes: inclined compression failure and inclined shear failure. With the decrease in shear span/depth ratio, the shear effect along the cylinder surface is increasingly apparent, and the failure mode transforms from diagonal compression failure to diagonal shear failure. At the same time, the effects of longitudinal reinforcement and stirrup gradually decrease.(2)The shear span/depth ratio exerts a significant influence on the shear performance of steel-fiber-reinforced high-strength concrete corbels, and the bearing capacity decreases obviously with an increase in the shear span/depth ratio. For a given shear span/depth ratio, the shear capacity of a corbel specimen under concentrated loads is greatly affected by the longitudinal reinforcement ratio, followed by the stirrup reinforcement ratio and the steel fiber content.(3)Under the same shear span/depth ratio, the cracking load increases with the increase in steel fiber content. Moreover, the second influence results from the stirrup reinforcement ratio, followed by the longitudinal reinforcement ratio.(4)The calculation values of the shear capacity of steel fiber high-strength concrete corbel specimens with GB 50010-2010 are close to the test values. It is important to note that the empirical formula used in GB 50010-2010 is based on ordinary concrete corbel specimens. While the prediction results of the ACI 318-19 code, EN 1992-1-1:2004 code, and CSA A23.3-19 code using the strut-and-tie model are more conservative, it is generally considered as an alternative design scheme.(5)It is worth noting that these conclusions are based on the test results of multivariate finite specimens. Further research is needed to thoroughly study corbels with different steel fiber content.

## Figures and Tables

**Figure 1 materials-16-03055-f001:**
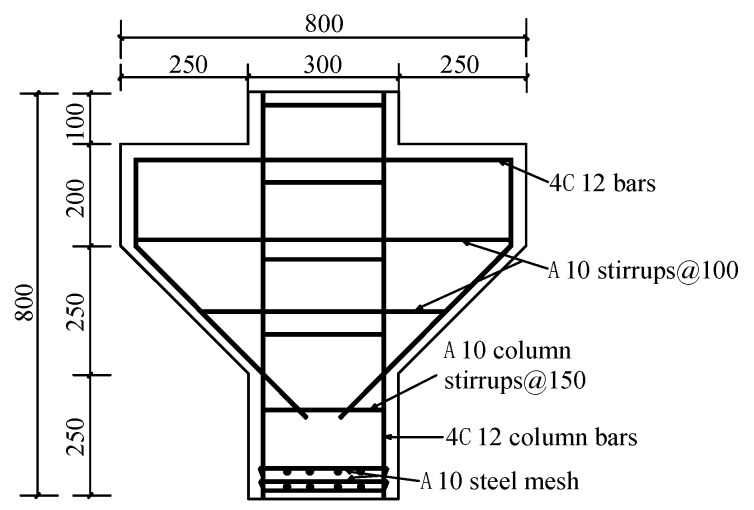
Dimensions of specimens and arrangement of reinforcement.

**Figure 2 materials-16-03055-f002:**
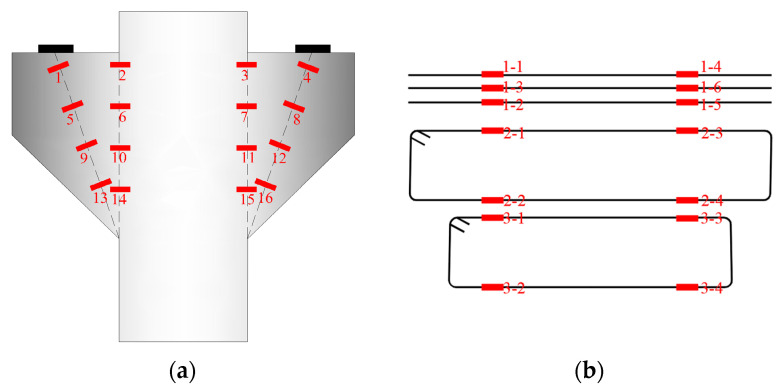
Strain gauge layout of concrete and steel. (**a**) Strain gauge layout of concrete. (**b**) Strain gauge layout of steel.

**Figure 3 materials-16-03055-f003:**
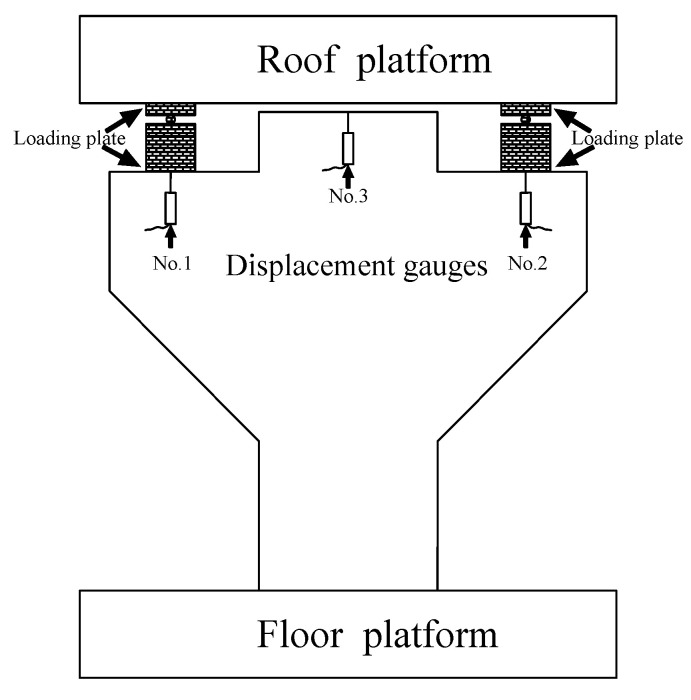
Displacement meter layout.

**Figure 4 materials-16-03055-f004:**
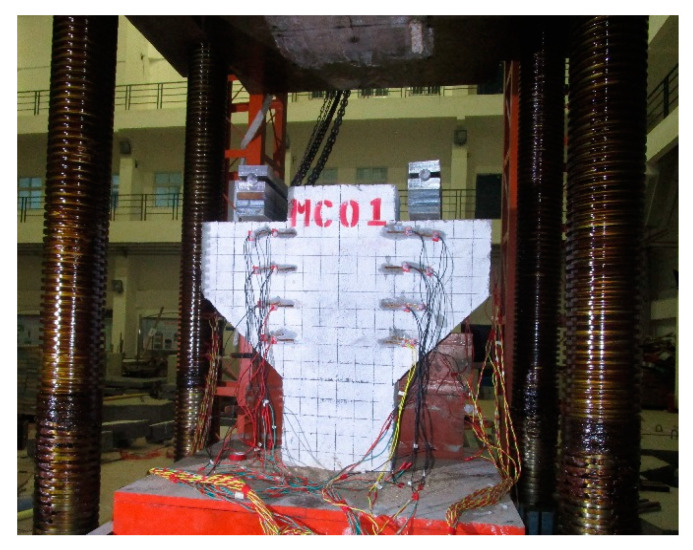
Test loading device for corbel specimens.

**Figure 5 materials-16-03055-f005:**
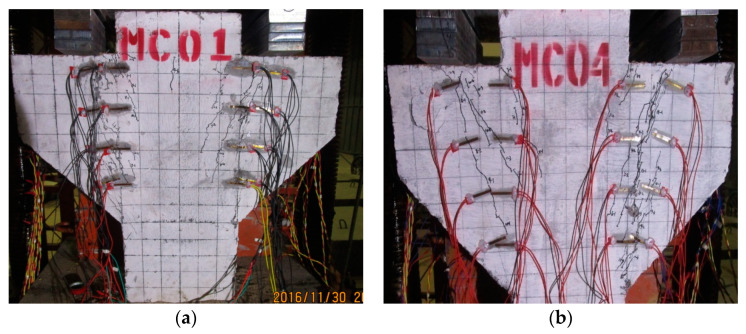
Failure pattern of typical specimens. (**a**) Diagonal shear failure (MC01). (**b**) Diagonal compression failure (MC04).

**Figure 6 materials-16-03055-f006:**
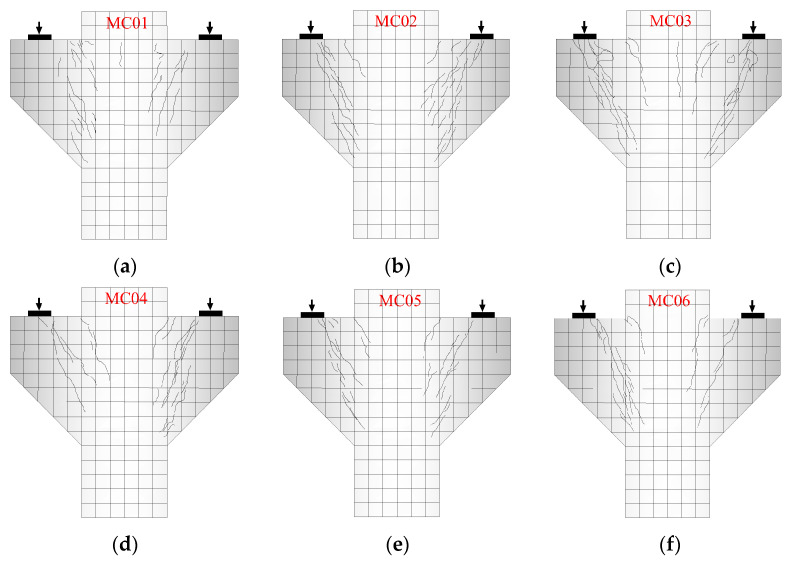
The crack failure of the specimens. (**a**) MC01; (**b**) MC02; (**c**) MC03; (**d**) MC04; (**e**) MC05; (**f**) MC06; (**g**) MC07; (**h**) MC08; (**i**) MC09.

**Figure 7 materials-16-03055-f007:**
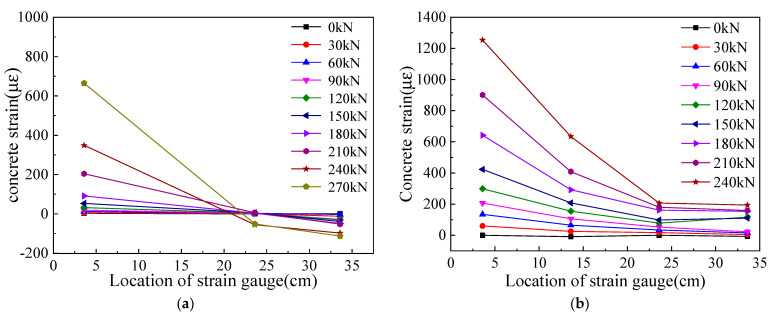
Concrete strain distribution of corbel. (**a**) Corbel normal section. (**b**) Corbel diagonal section.

**Figure 8 materials-16-03055-f008:**
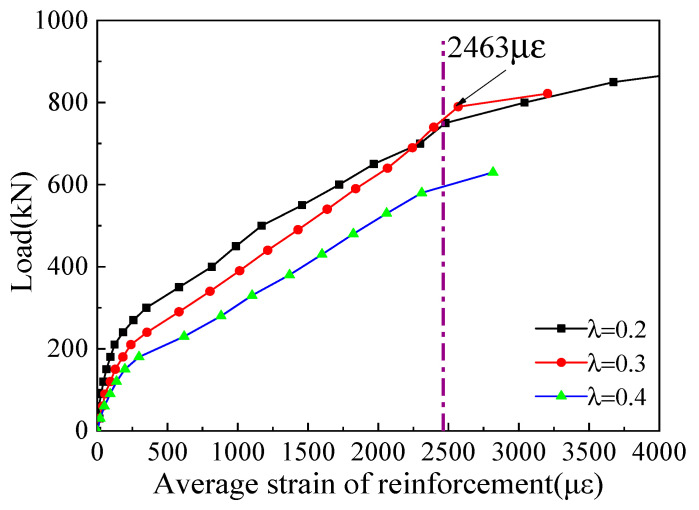
Relationship between load and average strain of longitudinal reinforcement under different λ.

**Figure 9 materials-16-03055-f009:**
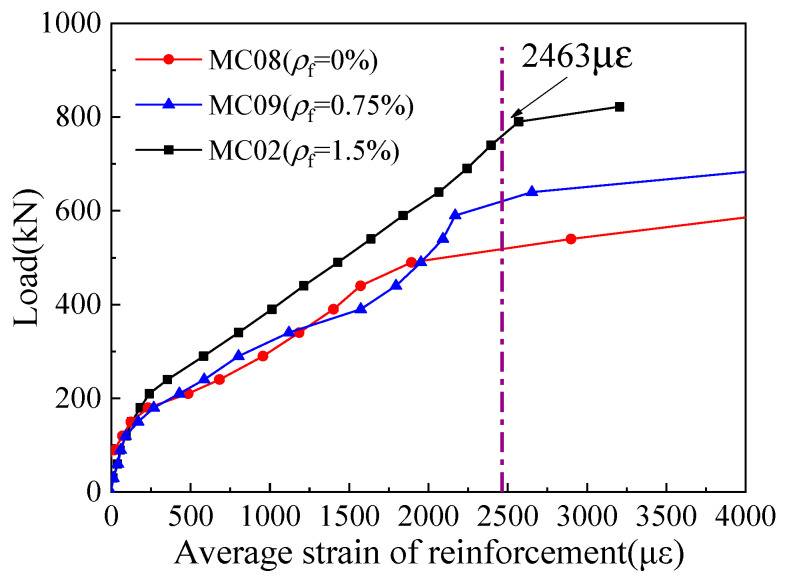
Relationship between load and average strain of longitudinal reinforcement under different *ρ*_f_.

**Figure 10 materials-16-03055-f010:**
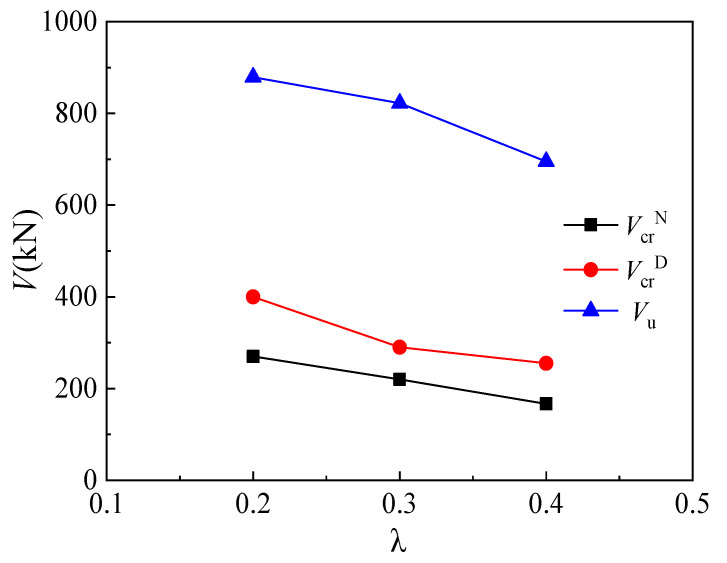
Relationship between load and shear span/depth ratio.

**Figure 11 materials-16-03055-f011:**
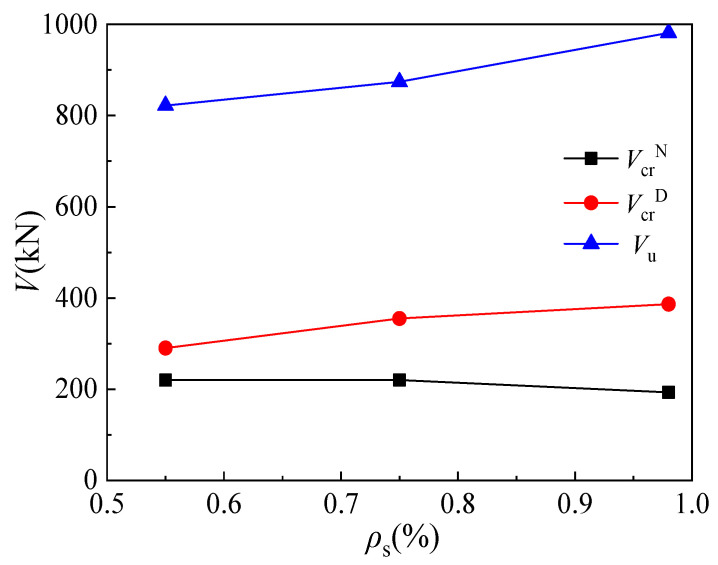
Relationship between load and longitudinal reinforcement ratio.

**Figure 12 materials-16-03055-f012:**
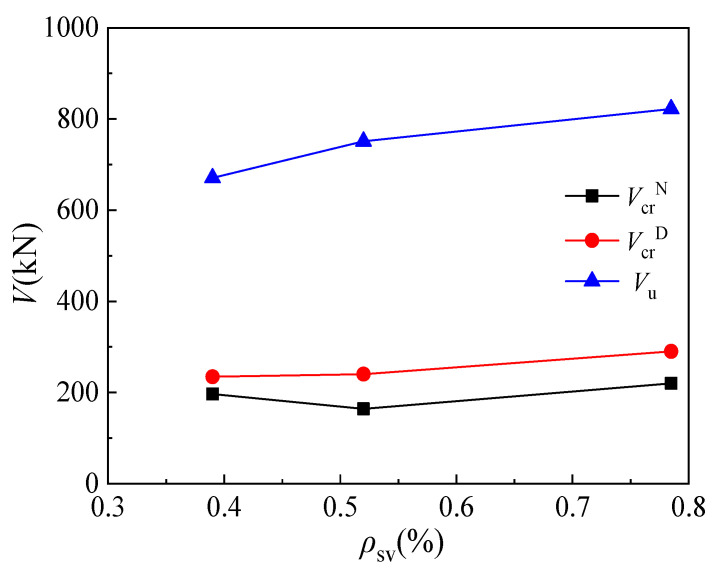
Relationship between load and stirrup reinforcement ratio.

**Figure 13 materials-16-03055-f013:**
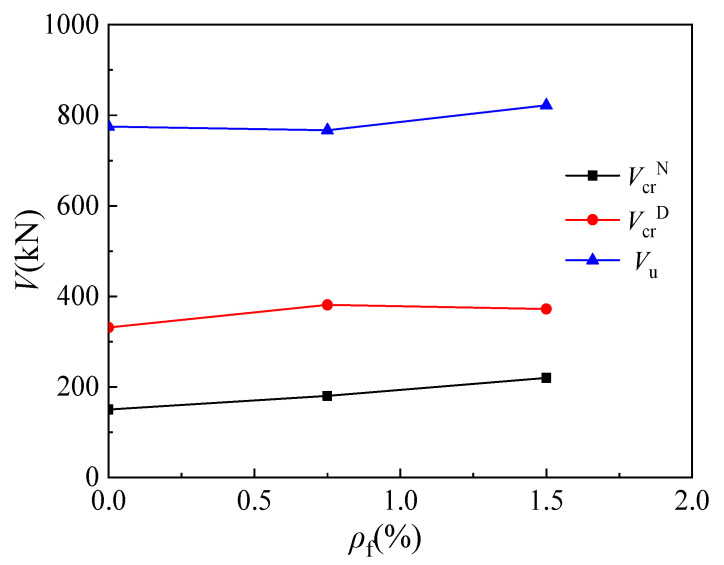
Relationship between load and steel fiber content.

**Figure 14 materials-16-03055-f014:**
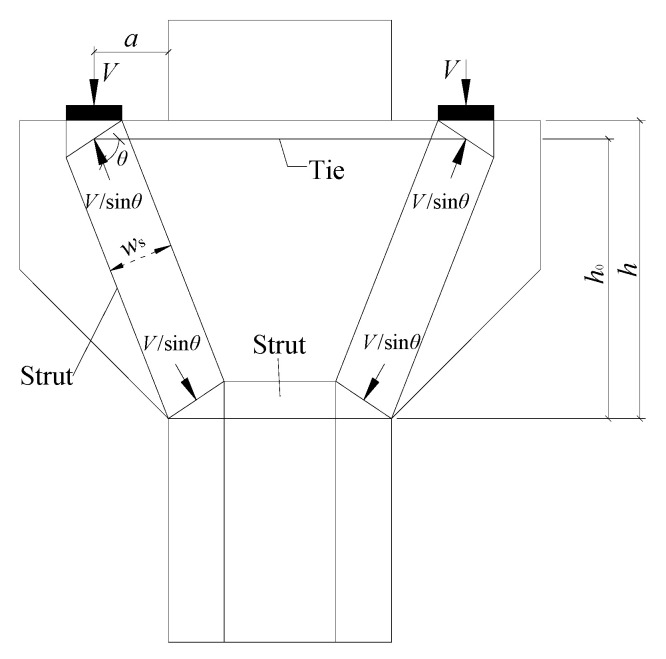
Simplified strut-and-tie model of a corbel.

**Table 1 materials-16-03055-t001:** Parameters of corbel specimens.

Specimen Number	Shear Span/Depth Ratio λ	Longitudinal Reinforcement Ratio *ρ*_s_ (%)	Stirrup Reinforcement Ratio *ρ*_sh_ (%)	Steel Fiber Content *ρ*_f_ (%)	*f*_y_ /MPa	*f*_ys_ /MPa	*f*_cu_ /MPa	*f*_t_ /MPa
MC01	0.2	0.55	0.785	1.5	425.2	333.7	72.6	3.6
MC02	0.3	0.55	0.785	1.5	425.2	333.7	72.6	3.6
MC03	0.4	0.55	0.785	1.5	425.2	333.7	72.6	3.6
MC04	0.3	0.75	0.785	1.5	425.2	333.7	72.6	3.6
MC05	0.3	0.98	0.785	1.5	425.2	333.7	72.6	3.6
MC06	0.3	0.55	0.39	1.5	425.2	333.7	72.6	3.6
MC07	0.3	0.55	0.52	1.5	425.2	333.7	72.6	3.6
MC08	0.3	0.55	0.785	0	425.2	333.7	73.2	4
MC09	0.3	0.55	0.785	0.75	425.2	333.7	69.8	3.3

*f*_y_ is the yield strength of HRB400 reinforcement with diameter of 12; *f*_ys_ is the yield strength of HPB300 reinforcement with diameter of 10; *f*_cu_ is the concrete’s compressive strength; *f*_t_ is the steel fiber concrete’s tensile strength.

**Table 2 materials-16-03055-t002:** Mixing proportions of C60 steel fiber concrete.

Number	Water (kg)	Cement (kg)	Sand (kg)	Stone (kg)	Superplasticizer (kg)	Steel Fiber (kg)
MC01	C60	573.3	681.9	1022.8	8.6	117
MC02	C60	573.3	681.9	1022.8	8.6	117
MC03	C60	573.3	681.9	1022.8	8.6	117
MC04	C60	573.3	681.9	1022.8	8.6	117
MC05	C60	573.3	681.9	1022.8	8.6	117
MC06	C60	573.3	681.9	1022.8	8.6	117
MC07	C60	573.3	681.9	1022.8	8.6	117
MC08	C60	500	612	1188	7.5	0
MC09	C60	533.35	702.35	1054	8	58.5

**Table 3 materials-16-03055-t003:** Material properties of steel fiber concrete.

Steel Fiber Content *ρ*_f_/%	Concrete Strength (MPa)	*f*_cu_ (MPa)	*f*_c_ (MPa)	*E*_c_ (GPa)
0	C60	73.2	55.9	38.7
0.75	C60	69.8	51.7	37.6
1.5	C60	72.6	49.8	37.2

**Table 4 materials-16-03055-t004:** Comparisons between experimental and calculation results.

Specimen Number (λ)	Test Values *V*_test_ (kN)	Calculated Values *V*_c_(kN)				*V*_test_/*V*_c_			
		GB 50010-2010	ACI318- 19	EN 1992-1- 1:2004	CSA A23.3 -19	GB 50010-2010	ACI318- 19	EN 1992-1- 1:2004	CSA A23.3 -19
MC01 (0.2)	879.0	716.40	456.38	479.19	473.67	1.227	1.926	1.834	1.856
MC02 (0.3)	822.0	477.60	423.22	444.37	417.22	1.721	1.942	1.850	1.970
MC03 (0.4)	695.0	358.20	390.33	409.83	363.60	1.940	1.781	1.696	1.911
MC04 (0.3)	874.0	648.53	423.22	444.37	417.22	1.348	2.065	1.967	2.095
MC05 (0.3)	981.5	844.89	423.22	444.37	417.22	1.162	2.319	2.209	2.352
MC06 (0.3)	670.5	477.60	423.22	444.37	417.22	1.404	1.584	1.509	1.607
MC07 (0.3)	751.0	477.60	423.22	424.54	417.22	1.572	1.774	1.690	1.800
MC08 (0.3)	775.0	477.60	404.82	424.54	399.08	1.623	1.914	1.826	1.942
MC09 (0.3)	767.0	477.60	427.22	448.62	421.16	1.606	1.795	1.710	1.821
Mean						1.511	1.900	1.810	1.928
Square						0.055	0.038	0.035	0.038

## Data Availability

Data are contained within the article.

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
