# Peer review of "Experimental Investigation on Shear Capacity of Steel-Fiber-Reinforced High-Strength Concrete Corbels"

_materials, 2023, doi:10.3390/ma16083055_

Round 1

Reviewer 1 Report

1- Materials should be given in details.

2- Shear failure mechanism of corbels should be given in photos and crack failure discussion should refer to these photos.

3- Failure modes should be discussed in a separate section without including loading and strain values. 

4- in the section "4" it was mentioned that "The shear capacity of steel fiber reinforced concrete corbels is influenced by several interrelated factors, including steel fiber content ratio, shear span-depth ratio, longitudinal reinforcement ratio, concrete strength, and stirrup reinforcement ratio. This study primarily focuses on analyzing the impact of shear span-depth ratio, longitudinal reinforcement ratio, stirrup reinforcement ratio, and steel fiber content ratio". I do not agree with the authors in this regard and what they focused on has been already highlighted in previous studies.

5- Table 1 reveals that the testing results do not comply with the codes' results! this conclusion is poor.

6- No need to discuss the results in conclusions.  

Reviewer 2 Report

The topic is interesting and the increase of database on corbels withs steel reinforcement fibers deserves for pubblication. Some aspects should be addressed:

1) In Table 1. Nomenclature of specimens: MC09 is repeated and MC08 does not appear.

2) Please, define "diagonal cracking load".  How is the diagonal crac king loadidentified in your tests.

Reviewer 3 Report

1.      In the “Introduction” section, the authors shall explain the problem statement of this research, motivations, research gap, novelty, and implications of this study.

2.      The current study lacks the section “Literature review” to discuss the advantages of having ductile members in earthquake-resisting structures, also up-to-date practices to improve structural ductility should be addressed. The authors may consult (but are not limited to) the following references:

-        https://doi.org/10.14359/51700951

-        https://doi.org/10.1061/(ASCE)CC.1943-5614.0001218

-        https://doi.org/10.14359/51737144

3.      In Figure 2, you should show the location of the applied load from the machine. You should also mention in the figure caption that the units are in millimeters.

4.      For Table 1, you should add one more column to report the tensile strength of Fiber Reinforced Concrete.

5.      You should provide information about the geometric and mechanical properties of steel fibers.

6.      Add new figures to show the strain gauge layout of concrete and steel.

7.      For the y-axis in figures 6 and 7, is the load equal to the measured load from the machine or equal to half of the reported force from the machine?

8.      Add a new section to discuss the limitations and scope of this study.

9.      Add a new section to explain the feasibility of this work and its potential impact on practice. 

10.   For the section “Conclusions”, you should explain the problem statement, merits, motivations, and approach of this study.  The implications and feasibility of this work should also be discussed.

11.   The technical writing in this manuscript needs more improvements, the authors shall improve the writing of this manuscript considering the flow and coherence of discussions. 

Round 2

Reviewer 3 Report

The authors have addressed the suggested comments.